# Accelerating Training of Transformer-Based Language Models with Progressive Layer Dropping

**Minjia Zhang**     **Yuxiong He**
Microsoft Corporation
{minjiaz,yuxhe}@microsoft.com

## Abstract

Recently, Transformer-based language models have demonstrated remarkable performance across many NLP domains. However, the unsupervised pre-training step of these models suffers from unbearable overall computational expenses. Current methods for accelerating the pre-training either rely on massive parallelism with advanced hardware or are not applicable to language modeling. In this work, we propose a method based on *progressive layer dropping* that speeds the training of Transformer-based language models, not at the cost of excessive hardware resources but from model architecture change and training technique boosted efficiency. Extensive experiments on BERT show that the proposed method achieves a 24% time reduction on average per sample and allows the pre-training to be $2.5\times$ faster than the baseline to get a similar accuracy on downstream tasks. While being faster, our pre-trained models are equipped with strong knowledge transferability, achieving comparable and sometimes higher GLUE score than the baseline when pre-trained with the same number of samples.

## 1   Introduction

Natural language processing (NLP) tasks, such as natural language inference [1, 2] and question answering [3–5], have achieved great success with the development of neural networks. It has been demonstrated recently that Transformer-based networks have obtained superior performance in many NLP tasks (e.g., the GLUE benchmark [6] and the challenging multi-hop reasoning task [7]) than recurrent neural networks or convolutional neural networks. BERT trains a deep bidirectional Transformer and obtains outstanding results with transfer learning [3]. RoBERTa [2], which is a robustly optimized version of BERT trained with more steps and larger corpora, achieves state-of-the-art results on 9 GLUE tasks. Megatron-LM [8] further advances the state-of-the-art in NLP by significantly increasing the size of BERT model. Finally, there are multiple research proposing different enhanced versions of Transformer-based networks, such as GPT-2/3 [9, 10], XLNet [1], SpanBERT [11], BioBERT [12], UniLM [13], Turing-NLG [14], and T5 [15]. Due to the exciting prospect, pre-training Transformer networks with a large corpus of text followed by fine-tuning on specific tasks has become a new paradigm for natural language processing.

Despite great success, a big challenge of Transformer networks comes from the training efficiency – even with self-attention and parallelizable recurrence [16], and extremely high performance hardware [17], the pre-training step still takes a significant amount of time. To address this challenge, mixed-precision training is explored [8, 18], where the forward pass and backward pass are computed in half-precision and parameter update is in single precision. However, it requires Tensor Cores [19], which do not exist in all hardware. Some work resort to distributed training [20, 21, 8]. However, distributed training uses large mini-batch sizes to increase the parallelism, where the training often converges to sharp local minima with poor generalizability even with significant hyperparameter tuning [22]. Subsequently, Yang et al. propose a layer-wise adaptive large batch optimizer called LAMB [23], allowing to train BERT with 32K batch size on 1024 TPU chips. However, this type of

approach often requires dedicated clusters with hundreds or even thousands of GPUs and sophisticated system techniques at managing and tuning distributed training, not to mention that the amount of computational resources is intractable for most research labs or individual practitioners.

In this paper, we speedup pre-training Transformer networks by exploring architectural change and training techniques, not at the cost of excessive hardware resources. Given that the training cost grows linearly with the number of Transformer layers, one straightforward idea to reduce the computation cost is to reduce the depth of the Transformer networks. However, this is restrictive as it often results in lower accuracy in downstream tasks compared to full model pre-training, presumably because of having smaller model capacities [24, 25]. Techniques such as Stochastic Depth have been demonstrated to be useful in accelerating supervised training in the image recognition domain [26]. However, we observe that stochastically removing Transformer layers destabilizes the performance and easily results in severe consequences such as model divergence or convergence to bad/suspicious local optima. Why are Transformer networks difficult to train with stochastic depth? Moreover, can we speed up the (unsupervised) pre-training of Transformer networks without hurting downstream performance?

To address the above challenges, we propose to accelerate pre-training of Transformer networks by making the following contributions. (i) We conduct a comprehensive analysis to answer the question: what makes Transformer networks difficult to train with stochastic depth. We find that both the choice of Transformer architecture as well as training dynamics would have a big impact on layer dropping. (ii) We propose a new architecture unit, called the *Switchable-Transformer* (ST) block, that not only allows switching on/off a Transformer layer for only a set portion of the training schedule, excluding them from both forward and backward pass but also stabilizes Transformer network training. (iii) We further propose a *progressive schedule* to add extra-stableness for pre-training Transformer networks with layer dropping – our schedule smoothly increases the layer dropping rate for each mini-batch as training evolves by adapting in time the parameter of the Bernoulli distribution used for sampling. Within each gradient update, we distribute a global layer dropping rate across all the ST blocks to favor different layers. (iv) We use BERT as an example, and we conduct extensive experiments to show that the proposed method not only allows to train BERT 24% faster than the baseline under the same number of samples but also allows the pre-training to be $2.5\times$ faster to get similar accuracy on downstream tasks. Furthermore, we evaluate the generalizability of models pre-trained with the same number of samples as the baseline, and we observe that while faster to train, our approach achieves a 1.1% higher GLUE score than the baseline, indicating a strong knowledge transferability.

## 2  Background and Related Work

Pre-training with Transformer-based architectures like BERT [3] has been demonstrated as an effective strategy for language representation learning [2, 1, 27, 8]. The approach provides a better model initialization for downstream tasks by training on large-scale unlabeled corpora, which often leads to a better generalization performance on the target task through fine-tuning on small data. Consider BERT, which consists a stack of $L$ Transformer layers [16]. Each Transformer layer encodes the the input of the i-th Transformer layer $x_i$ with $h_i = f_{LN}(x_i + f_{S-ATTN}(x_i))$, which is a multi-head self-attention sub-layer $f_{ATTN}$, and then by $x_{i+1} = f_{LN}(h_i + f_{FFN}(h_i))$, which is a feed-forward network $f_{FFN}$, where $x_{i+1}$ is the output of the i-th Transformer layer. Both sub-layers have an AddNorm operation that consists a residual connection [28] and a layer normalization ($f_{LN}$) [29]. The BERT model recursively applies the transformer block to the input to get the output.

While the Transformer-based architecture has achieved breakthrough results in modeling sequences for unsupervised language modeling [3, 9], previous work has also highlighted the training difficulties and excessively long training time [2]. To speed up the pre-training, ELECTRA [30] explores the adversarial training scheme by replacing masked tokens with alternatives sampled from a generator framework and training a discriminator to predict the replaced token. This increases the relative per-step cost, but leads to fewer steps, leading to the overall reduced costs. Another line of work focus on reducing the per-step cost. Since the total number of floating-point operations (FLOPS) of the forward and backward passes in the BERT pre-training process is linearly proportional to the depth of the Transformer blocks, reducing the number of Transformer layers brings opportunities to significantly speed up BERT pre-training. To show this, we plot the FLOPS per training iteration in Fig. 6, assuming we can remove a fraction of layers at each step. Each line in the figure shows the FLOPS using different layer removal schedules. Regardless of which schedule to choose, the

majority of FLOPS are reduced in the later steps, with the rate of keep probability saturating to a fixed value $\bar{\theta}$ (e.g., 0.5). We will describe our schedule in Section 4.2.

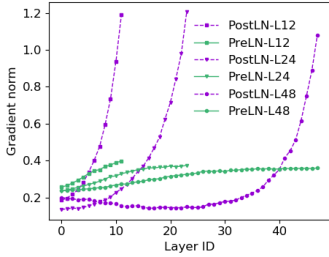

Figure 1: The norm of the gradient with respect to the weights, with PostLN and PreLN.

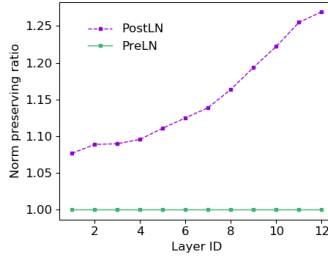

Figure 2: The norm preserving ratio with respect to the inputs, with PostLN and PreLN.

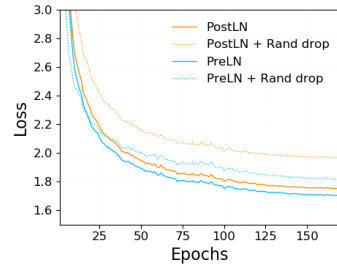

Figure 3: Lesioning analysis with PostLN and PreLN.

Despite the FLOPS reduction, directly training models like BERT with a smaller depth incurs a significant loss in accuracy even with knowledge distillation [24, 25]. Prior work [31] proposes to accelerate pre-training by first training a 3-layer BERT model and then growing the network depth to 6-layer and subsequently 12-layer. However, the number of steps required at each depth before the network growth is not known a prior, making applying this approach challenging in practice. On the other hand, stochastic depth has been successfully demonstrated to train deep models with reduced expected depth [26, 32]. However, we observe that directly pre-training BERT with randomly dropping $f_{ATTN}$ and $f_{FFN}$ converges to bad/suspicious local optima under the same hyperparameter setting. When increasing the learning rate, the training often diverges even by tuning the warmup ratio. What causes the instability of BERT pre-training with layer drop?

## 3  Preliminary Analysis

This section presents several studies that guided the design of the approach introduced in Section 4. We used BERT trained on Bookcorpus and Wikipedia dataset from Devlin et. al. with standard settings as the baseline [1]. First, we carry out a comparison between BERT with PostLN and PreLN. Our goal is to measure how effective these two methods at stabilizing BERT training. Our second analysis considers measuring the dynamics of BERT pre-training, including both spatial and temporal dimensions. Finally, we analyze the effect of the removal of the Transformer layers. This leads us to identify appealing choices for our target operating points.

### 3.1  Training Stability: PostLN or PreLN?

We consider two variants of BERT, namely the PostLN and PreLN. The default BERT employs PostLN, with layer normalization applied after the addition in Transformer blocks. The PreLN changes the placement of the location of $f_{LN}$ by placing it only on the input stream of the sublayers so that $h_i = x_i + f_{S-ATTN}(f_{LN}(x_i))$ and then $x_{i+1} = h_i + f_{FFN}(f_{LN}(h_i))$, which is a modification described by several recent works to establish identity mapping for neural machine translation [33–37]. Fig. 1 reports the norm of gradients with respect to weights in backward propagation for both methods, varying the depth $L$ (e.g., 12, 24, 48). The plot shows that while PostLN suffers from unbalanced gradients (e.g., vanishing gradients as the layer ID decreases), PreLN eliminates the unbalanced gradient problem (solid green lines) and the gradient norm stays almost same for any layer. Furthermore, Fig. 2 shows that for PreLN the gradients with respect to input $x_i$ have very similar magnitudes (norm preserving ratio close to 1) at different layers, which is consistent with prior findings that a neural model should preserve the gradient norm between layers so as to have well-conditioning and faster convergence [38, 39]. Indeed, we find that PostLN is more sensitive to the choice of hyperparameters, and training often diverges with more aggressive learning rates (more results in Section 5), whereas PreLN avoids vanishing gradients and leads to more stable optimization. We also provide preliminary theoretical results in Appendix B on why PreLN is beneficial.

## 3.2 Corroboration of Training Dynamics

Hereafter we investigate the representation $x_i$ learned at different phases of BERT pre-training and at different layers. Fig. 4 shows the L2 norm distances and cosine similarity, which measures the angle between two vectors and ignores their norms, between the input and output embeddings, with PostLN and PreLN, respectively. We draw several observations.

First, the dissimilarity (Fig. 4a and Fig. 4b) stays high for both PostLN and PreLN at those higher layers in the beginning, and the L2 and cosine similarity seems to be less correlated (e.g., step = 300). This is presumably because, at the beginning of the training, the model weights are randomly initialized, and the network is still actively adjusting weights to derive richer features from input data. Since the model is still positively self-organizing on the network parameters toward their optimal configuration, dropping layers at this stage is not an interesting strategy, because it can create inputs with large noise and disturb the positive co-adaption process.

Second, as the training proceeds (Fig. 4c and Fig. 4d), although the dissimilarity remains relatively high and bumpy for PostLN, the similarity from PreLN starts to increase over successive layers, indicating that while PostLN is still trying to produce new representations that are very different across layers, the dissimilarity from PreLN is getting close to zero for upper layers, indicating that the upper layers are getting similar estimations. This can be viewed as doing an unrolled iterative refinement [40], where a group of successive layers iteratively refine their estimates of the same representations instead of computing an entirely new representation. Although the viewpoint was originally proposed to explain ResNet, we demonstrate that it is also true for language modeling and Transformer-based networks. Appendix C provides additional analysis on how PreLN provides extra preservation of feature identity through unrolled iterative refinement.

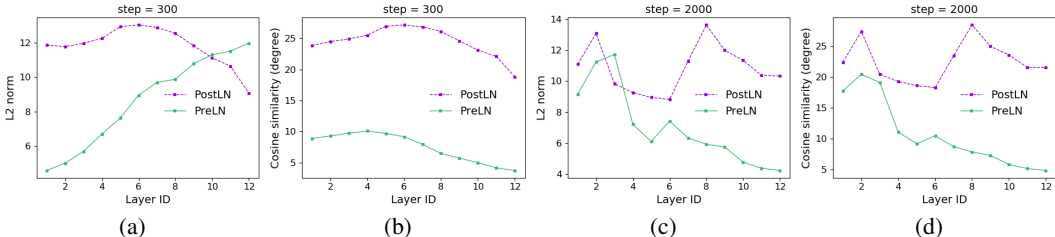

(a)            (b)            (c)            (d)

Figure 4: The L2 distance and cosine similarity of the input and output embeddings for BERT with PostLN and PreLN, at different layers and different steps. We plot the inverse of cosine similarity (arccosine) in degrees, so that for both L2 and arccosine, the lower the more similar.

## 3.3 Effect of Lesioning

We randomly drop layers with a keep ratio $\theta = 0.5$ to test if dropping layers would break the training because dropping any layer changes the input distribution of all subsequent layers. The results are shown in Fig. 3. As shown, removing layers in PostLN significantly reduces performance. Moreover, when increasing the learning rate, it results in diverged training. In contrast, this is not the case for PreLN. Given that later layers in PreLN tend to refine an estimate of the representation, the model with PreLN has less dependence on the downsampling individual layers. As a result, removing Transformer layers with PreLN has a modest impact on performance (slightly worse validation loss at the same number of training samples). However, the change is much smaller than with PostLN. It further indicates that if we remove layers, especially those higher ones, it should have only a mild effect on the final result because doing so does not change the overall estimation the next layer receives, only its quality. The following layers can still perform mostly the same operation, even with some relatively little noisy input. Furthermore, as Fig. 4 indicates, since the lower layers remain to have a relatively high dissimilarity (deriving new features), they should be less frequently dropped. Overall, these results show that, to some extent, the structure of a Transformer network with PreLN can be changed at runtime without significantly affecting performance.

# 4 Our Approach: Progressive Layer Dropping

This section describes our approach, namely progressive layer dropping (PLD), to accelerate the pre-training of Transformer-based models. We first present the Switchable-Transformer blocks, a new unit that allows us to train models like BERT with layer drop and improved stability. Then we introduce the progressive layer drop procedure.

## 4.1 Switchable-Transformer Blocks

In this work, we propose a novel transformer unit, which we call "Switchable-Transformer " (ST) block. Compared with the original Transformer block (Fig. 5a), it contains two changes.

**Identity mapping reordering.** The first change is to establish identity mapping within a transformer block by placing the layer normalization only on the input stream of the sublayers (i.e., use PreLN to replace PostLN) (Fig. 5b) for the stability reason described in Section 3.1.

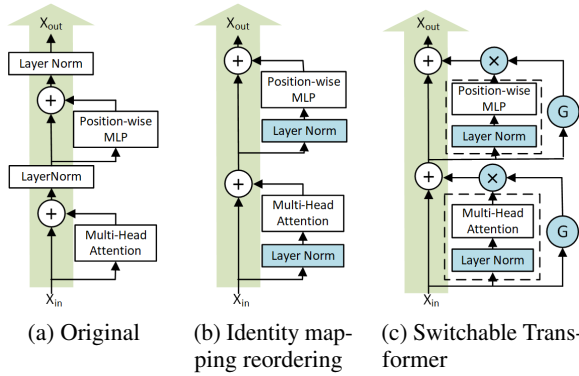

(a) Original    (b) Identity mapping reordering    (c) Switchable Transformer

Figure 5: Transformer variants, showing a single layer block.

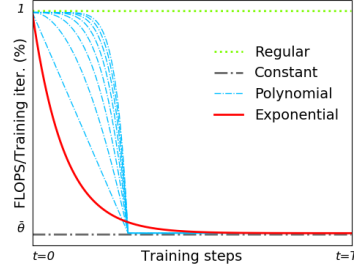

Figure 6: FLOPS per training iteration normalized to the baseline.

**Switchable gates.** Next, we extend the architecture to include a gate for each sub-layer (Fig. 5c), which controls whether a sub-layer is disabled or not during training. In particular, for each mini-batch, the two gates for the two sublayers decide whether to remove their corresponding transformation functions and only keep the identify mapping connection, which is equivalent to applying a conditional gate function $G$ to each sub-layer as follows:

$$
h_i = x_i + G_i \times f_{S-ATTN}(f_{LN}(x_i)) \times \frac{1}{p_i}
$$
$$
x_{i+1} = h_i + G_i \times f_{FFN}(f_{LN}(h_i)) \times \frac{1}{p_i}
$$

(1)

In our design, the function $G_i$ only takes 0 or 1 as values, which is chosen randomly from a Bernoulli distribution (with two possible outcomes), $G_i \sim B(1, p_i)$, where $p_i$ is the probability of choosing 1. Because the blocks are selected with probability $p_i$ during training and are always presented during inference, we re-calibrate the layers' output by a scaling factor of $\frac{1}{p_i}$ whenever they are selected.

## 4.2 A Progressive Layer Dropping Schedule

Based on the insights from Section 3.2, and inspired by prior work on curriculum learning [41, 42] we propose a progressive schedule $\theta(t)$ – a temporal schedule for the expected number of ST blocks that are retained. We limit ourselves to monotonically decreasing functions so that the likelihood of layer dropping can only increase along the temporal dimension. We constrain $\theta(t)$ to be $\theta(t) \geq \bar{\theta}$ for any t, where $\bar{\theta}$ is a limit value, to be taken as $0.5 \leq \bar{\theta} \leq 0.9$ (Section 5). Based on this, we define the progressive schedule $\theta(t)$ as:

**Definition 4.1.** *A progressive schedule is a function* $t \rightarrow \theta(t)$ *such that* $\theta(0) = 1$ *and* $\lim_{t \rightarrow \infty} \theta(t) \rightarrow \bar{\theta}$, *where* $\bar{\theta} \in (0, 1]$.

**Progress along the time dimension.** Starting from the initial condition $\theta(0) = 1$ where no layer drop is performed, layer drop is gradually introduced. Eventually (i.e., when $t$ is sufficiently large), $\theta(t) \to \bar{\theta}$. According to Def. 4.1, in our work, we use the following schedule function:

$$\bar{\theta}(t) = (1 - \bar{\theta})exp(-\gamma \cdot t) + \bar{\theta}, \gamma > 0 \tag{2}$$

By considering Fig. 6, we provide intuitive and straightforward motivation for our choice. The blue curve in Fig. 6 are polynomials of increasing degree $\delta = \{1, .., 8\}$ (left to right). Despite fulfilling the initial constraint $\theta(0) = 1$, they have to be manually thresholded to impose $\theta(t) \to \bar{\theta}$ when $t \to \infty$, which introduces two more parameters ($\delta$ and the threshold). In contrast, in our schedule, we fix $\gamma$ using the following simple heuristics $\gamma = \frac{100}{T}$, since it implies $|\theta(T) - \bar{\theta}| < 10^{-5}$ for $\theta(t) \approx \bar{\theta}$ when $t \approx T$, and it is reasonable to assume that T is at the order of magnitude of $10^4$ or $10^5$ when training Transformer networks. In other words, this means that for a big portion of the training, we are dropping $(1 - \bar{\theta})$ ST blocks, accelerating the training efficiency.

**Distribution along the depth dimension.** The above progressive schedule assumes all gates in ST blocks take the same $p$ value at each step t. However, as shown in Fig. 4, the lower layers of the networks should be more reliably present. Therefore, we distribute the global $\bar{\theta}$ across the entire stack so that lower layers have lower drop probability linearly scaled by their depth according to equation 3. Furthermore, we let the sub-layers inside each block share the same schedule, so when $G_i = 1$, both the inner function $f_{ATTN}$ and $f_{FFN}$ are activated, while they are skipped when $G_i = 0$. Therefore, each gate has the following form during training:

$$p_l(t) = \frac{i}{L}(1 - \bar{\theta}(t)) \tag{3}$$

Combining Eqn. 3 and Eqn. 2, we have the progressive schedule for an ST block below.

$$\theta_i(t) = \frac{i}{L}(1 - (1 - \bar{\theta}(t))exp(-\gamma \cdot t) - \bar{\theta}(t)) \tag{4}$$

**Putting it together.** Note that because of the exist of the identity mapping, when an ST block is bypassed for a specific iteration, there is no need to perform forward-backward computation or gradient updates, and there will be updates with significantly shorter networks and more direct paths to individual layers. Based on this, we design a stochastic training algorithm based on ST blocks and the progressive layer-drop schedule to train models like BERT faster, which we call *progressive layer dropping* (Alg. 1). The expected network depth, denoted as $\bar{L}$, becomes a random variable. Its expectation is given by: $E(\bar{L}) = \sum_{t=0}^{T} \sum_{i=1}^{L} \theta(i, t)$. With $\bar{\theta} = 0.5$, the expected number of ST blocks during training reduces to $E(\bar{L}) = (3L-1)/4$ or $E(\bar{L}) \approx 3L/4$ when T is large. For the 12-layer BERT model with L=12 used in our experiments, we have $E(\bar{L}) \approx 9$. In other words, with progressive layer dropping, we train BERT with an average number of 9 layers. This significantly improves the pre-training speed of the BERT model. Following the calculations above, approximately 25% of

---

**Algorithm 1    Progressive_Layer_Dropping**

1: **Input:** $keep\_prob\ \bar{\theta}$
2: InitBERT($switchable\_transformer\_block$)
3: $\gamma \leftarrow \frac{100}{T}$
4: **for** t $\leftarrow$ 1 to T **do**
5:      $\theta_t \leftarrow (1 - \bar{\theta})exp(-\gamma \cdot t) + \bar{\theta}$
6:      step $\leftarrow \frac{1-\theta_t}{L}$
7:      $p \leftarrow 1$
8:      **for** l $\leftarrow$ 1 to L **do**
9:          action $\sim$ Bernoulli(p)
10:          **if** action == 0 **then**
11:              $x_{i+1} \leftarrow x_i$
12:          **else**
13:              $x_i' \leftarrow x_i + f_{ATTN}(f_{LN}(x_i)) \times \frac{1}{p}$
14:              $x_{i+1} \leftarrow x_i' + f_{FFN}(f_{LN}(x_i')) \times \frac{1}{p}$
15:          $x_i \leftarrow x_{i+1}$
16:          $p \leftarrow$ p - step
17:      Y $\leftarrow output\_layer(x_L)$
18:      loss $\leftarrow loss\_fn(\bar{Y}, Y)$
19:      backward(loss)

---

FLOPS could be saved under the drop schedule with $\bar{\theta} = 0.5$. We recover the model with full-depth blocks at fine-tuning and testing time.

## 5    Evaluation

We show that our method improves the pre-training efficiency of Transformer networks, and the trained models achieve competitive or even better performance compared to the baseline on transfer learning downstream tasks. We also show ablation studies to analyze the proposed training techniques.

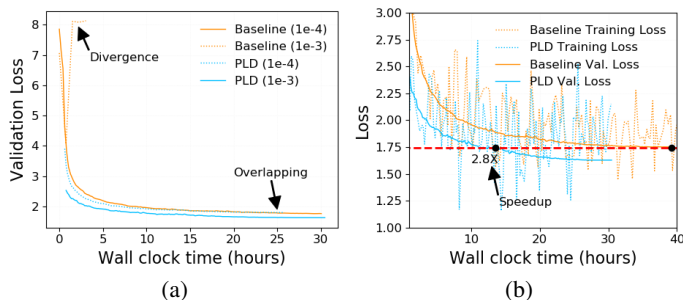

Table 1: Training time comparison. Sample RD standards for sample reduction. SPD represents speedup.

| | Training Time | Sample RD | SPD |
|---|---|---|---|
| Baseline ckp186 | 38.45h | 0 | 1 |
| PLD ckp186 | 29.22h | 0 | 1.3× |
| PLD ckp100 | 15.56h | 46% | 2.5× |
| PLD ckp87 | 13.53h | 53% | 2.8× |

Figure 7: The convergence curve of the baseline and our proposed method regarding the wall-clock time.

**Datasets.**   We follow Devlin et al. [3] to use English Wikipedia corpus and BookCorpus for pre-training. By concatenating the two datasets, we obtain our corpus with roughly 2.8B word tokens in total, which is comparable with the data corpus used in Devlin et al. [3].   We segment documents into sentences with 128 tokens; We normalize, lower-case, and tokenize texts using Wordpiece tokenizer [3]. The final vocabulary contains 30,528 word pieces. We split documents into one training set and one validation set (300:1). For fine-tuning, we use GLUE (General Language Understanding Evaluation), a collection of 9 sentence or sentence-pair natural language understanding tasks including question answering, sentiment analysis, and textual entailment. It is designed to favor sample-efficient learning and knowledge-transfer across a range of different linguistic tasks in different domains.

**Training details.**   We use our own implementation of the BERT model [3] based on the Hugging-face[1] PyTorch implementation. All experiments are performed on 4×DGX-2 boxes with 64×V100 GPUs. Data parallelism is handled via PyTorch DDP (Distributed Data Parallel) library [43]. We recognize and eliminate additional computation overhead: we overlap data loading with computation through the asynchronous prefetching queue; we optimize the BERT output processing through sparse computation on masked tokens. Using our pre-processed data, we train a 12-layer BERT-base model from scratch as the baseline. We use a warm-up ratio of 0.02 with $lr_{max}$=1e$^{-4}$. Following [3], we use Adam as the optimizer. We train with batch size 4K for 200K steps, which is approximately 186 epochs. The detailed parameter settings are listed in the Appendix A. We fine-tune GLUE tasks for 5 epochs and report the median development set results for each task over five random initializations.

## 5.1   Experimental Results

**Pre-training convergence comparisons.**   Fig. 7a visualizes the convergence of validation loss regarding the computational time. We make the following observations. First, with $lr_{max}$=1e$^{-4}$, the convergence rate of our algorithm and the baseline is very close. This verifies empirically that our progressive layer dropping method does not hurt model convergence. Second, when using a larger learning rate $lr_{max}$=1e$^{-3}$, the baseline diverges. In contrast, our method shows a healthy convergence curve and is much faster. This confirms that our architectural changes stabilize training and allows BERT training with more aggressive learning rates.

**Speedup.**   Fig. 7b shows both the training curve (dotted) and the validation curve (solid) of the baseline and PLD with a zoomed-in view. The baseline curve becomes almost flat at epoch 186, getting a validation loss of 1.75. In contrast, PLD reaches the same validation loss at epoch 87, with 53% fewer training samples. Furthermore, PLD achieves a 24% time reduction when training the same number of samples. This is because our approach trains the model with a smaller number of expected depth for the same number of steps. It is slightly lower than the 25% GFLOPS reduction in the analysis because the output layer still takes a small amount of computation even after optimizations. The combination of these two factors, yields 2.8× speedup in end-to-end wall-clock training time over the baseline, as shown in Table 1.

**Downstream task accuracy.**   Despite improved training speed, one may still wonder whether such a method is as effective as the baseline model on downstream tasks. Table 2 shows our results on the GLUE dataset compared to the baseline. Our baseline is comparable with the original BERT-Base (on the test set), and our PLD method achieves a higher GLUE score than our baseline (83.2 vs. 82.1)

when fine-tuning the checkpoint (186). We also dump model checkpoints from different epochs during pre-training and fine-tune these models. The checkpoint 87 corresponds to the validation loss at 1.75 achieved by PLD. The GLUE score is slightly worse than the baseline (81.6 vs. 82.1). However, by fine-tuning at checkpoint 100, PLD achieves a higher score than the baseline (82.3 vs. 82.1) at checkpoint 186. In terms of the pre-training wall clock time, PLD requires 15.56h vs. the baseline with 39.15h to get similar accuracy on downstream tasks, which corresponds to a 2.5× speedup.

Table 2: The results on the GLUE benchmark. The number below each task denotes the number of training examples. The metrics for these tasks can be found in the GLUE paper [6]. We compute the geometric mean of the metrics as the GLUE score.

| Model | RTE (Acc.) | MRPC (F1/Acc.) | STS-B (PCC/SCC) | CoLA (MCC) | SST-2 (Acc.) | QNLI (Acc.) | QQP (F1/Acc.) | MNLI-mm -/m (Acc.) | GLUE |
|---|---|---|---|---|---|---|---|---|---|
| | 2.5K | 3.7K | 5.7K | 8.5K | 67K | 108K | 368K | 393K | |
| $BERT_{base}$ (original) | 66.4 | 88.9/84.8 | 87.1/89.2 | 52.1 | **93.5** | **90.5** | 71.2/89.2 | **84.6**/83.4 | 80.7 |
| $BERT_{base}$ (Baseline, ckp186) | 67.8 | 88.0/86.0 | 89.5/**89.2** | 52.5 | 91.2 | 87.1 | 89.0/90.6 | 82.5/83.4 | 82.1 |
| $BERT_{base}$ (PLD, ckp87) | 66 | 88.2/85.6 | 88.9/88.4 | 54.5 | 91 | 86.3 | 87.4/89.1 | 81.6/82.4 | 81.6 |
| $BERT_{base}$ (PLD, ckp100) | 68.2 | 88.2/85.8 | 89.3/88.9 | 56.1 | 91.5 | 86.9 | 87.7/89.3 | 82.4/82.6 | 82.3 |
| $BERT_{base}$ (PLD, ckp186) | **69** | **88.9/86.5** | **89.6**/89.1 | **59.4** | 91.8 | 88 | **89.4/90.9** | 83.1/**83.5** | **83.2** |

Fig. 8a and 8b illustrate the fine-tuning results between the baseline and PLD on GLUE tasks over different checkpoints (due to space limitations, the complete set of results are provided in Appendix D). In each figure, we observe that both curves have a similar shape at the beginning because no layer drop is added. For later checkpoints, PLD smoothly adds layer drop. Interestingly, we note that the baseline model has fluctuations in testing accuracy. In contrast, the downstream task accuracy from PLD is consistently increasing as the number of training epochs increases. This indicates that PLD takes a more robust optimization path toward the optimum. Furthermore, although PLD takes a shorter time to train, it generalizes competitively and sometimes better on downstream tasks than our baseline does. This is presumably because by selecting a different subset of layers in each mini-batch, PLD encourages the layers to produce good results independently. This allows the model to learn a more general representation through averaging the noise patterns and create the effect of ensembling different sub-networks during inference, which helps the model to generalize better. These results confirm the validity of PLD.

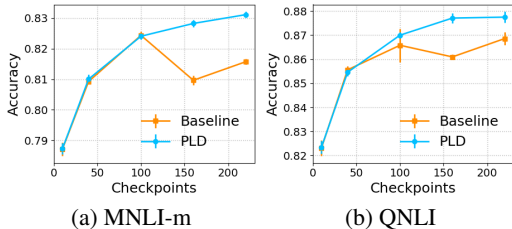

(a) MNLI-m          (b) QNLI

Figure 8: The fine-tuning accuracy results at different checkpoints.

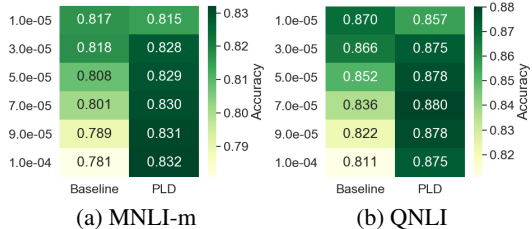

(a) MNLI-m          (b) QNLI

Figure 9: The fine-tuning accuracy results, varying different learning rates.

## 5.2 Ablation Studies

**Downstream task fine-tuning sensitivity.** To further verify that our approach not only stabilizes training but also improves downstream tasks, we show a grid search on learning rates {1e-5, 3e-5, 5e-5, 7e-5, 9e-5, 1e-4}. As illustrated in Fig. 9, the baseline is vulnerable to the choice of learning rates. Specifically, the fine-tuning results are often much worse with a large learning rate, while PLD is more robust and often achieves better results with large learning rates.

**The Effect of $\bar{\theta}$.** We test different values of the keep ratio $\bar{\theta}$ and identify $0.5 \leq \bar{\theta} \leq 0.9$ as a good range, as shown in Fig. 12 in the Appendix. We observe that the algorithm may diverge if $\bar{\theta}$ is too small (e.g., 0.3).

**PLD vs. PreLN.** To investigate the question on how PLD compares with PreLN, we run both PreLN with the hyperparameters used for training PostLN (lr=1e-4) and the hyperparameters used for PLD (lr=1e-3) to address the effect from the choice of hyperparameters. We train all configurations for the same number of epochs and fine-tune following the standard procedure. In both cases, PreLN

is 24% slower than PLD, because PreLN still needs to perform the full forward and backward propagation in each iteration.

Table 3 shows the fine-tuning results on GLUE tasks. When trained with the same hyperparameters as PostLN, PreLN appears to have a much worse GLUE score (80.2) compared with PostLN (82.1) on downstream tasks. This is because PreLN restricts layer outputs from depending too much on their own residual branches and inhibits the network from reaching its full potential, as recently studied in [44]. When trained with the large learning rate as PLD, PreLN's result have improved to 82.6 but is 0.6 points worse than PLD (83.2), despite using 24% more compute resource. PLD achieves better accuracy than PreLN because it encourages each residual branch to produce good results independently.

Table 3: Ablation studies of the fine-tuning results on the GLUE benchmark.

| Model | RTE (Acc.) | MRPC (F1/Acc.) | STS-B (PCC/SCC) | CoLA (MCC) | SST-2 (Acc.) | QNLI (Acc.) | QQP (F1/Acc.) | MNLI-m/mm (Acc.) | GLUE |
|---|---|---|---|---|---|---|---|---|---|
| BERT (Original) | 66.4 | **88.9**/84.8 | 87.1/89.2 | 52.1 | **93.5** | **90.5** | 71.2/89.2 | **84.6/83.4** | 80.7 |
| BERT + PostLN | 67.8 | 88.0/86.0 | 89.5/89.2 | 52.5 | 91.2 | 87.1 | 89.0/90.6 | 82.5/83.4 | 82.1 |
| BERT + PreLN + Same lr | 66.0 | 85.9/83.3 | 88.2/87.9 | 46.4 | 90.5 | 85.5 | 89.0/90.6 | 81.6/81.6 | 80.2 |
| BERT + PreLN + lr↑ | 67.8 | 86.7/84.5 | **89.6/89.1** | 54.6 | 91.9 | 88.1 | 89.3/**90.9** | **83.6/83.7** | 82.6 |
| Shallow BERT + PreLN + lr↑ | 66.0 | 85.9/83.5 | 89.5/88.9 | 54.7 | 91.8 | 86.1 | 89.0/90.6 | 82.7/82.9 | 81.8 |
| BERT + PreLN + lr↑ + Rand. | 68.2 | 88.2/86.2 | 89.3/88.8 | 56.8 | 91.5 | 87.2 | 88.6/90.3 | 82.9/83.3 | 82.7 |
| BERT + PreLN + lr↑ + TD | 68.2 | 88.6/**86.7** | 89.4/88.9 | 55.9 | 91.3 | 86.8 | 89.1/90.7 | 82.7/83.1 | 82.7 |
| BERT + PreLN + lr↑ + PLD | **69.0** | **88.9**/86.5 | **89.6/89.1** | **59.4** | 91.8 | 88.0 | **89.4/90.9** | 83.1/83.5 | **83.2** |

**PLD vs. Shallow network.** *Shallow BERT + PreLN + Large lr* in Table 3 shows the downstream task accuracy of the 9-layer BERT. Although having the same same number of training computational GFLOPS as ours, the shallow BERT underperforms PreLN by 0.8 points and is 1.4 points worse than PLD likely because the model capacity has been reduced by the loss of parameters.

**PLD vs. Random drop.** *BERT + PreLN + Large lr + Random* drops layers randomly with a fixed ratio (i.e., it has the same compute cost but without any schedule), similar to Stochastic Depth [26]. The GLUE score is 0.9 points better than shallow BERT under the same compute cost and 0.1 point better than PreLN while being 24% faster, indicating the strong regularization effect from stochastic depth. It is 0.5 points worse than PLD, presumably because a fixed ratio does not take into account the training dynamics of Transformer networks.

**Schedule impact analysis.** *BERT + PreLN + Large lr + TD only (32-bit\*)* disables the schedule along the depth dimension (DD) and enables only the schedule along the temporal dimension (TD) in training. Its GLUE score matches "Random", suggesting that the temporal schedule has similar performance as the fixed constant schedule along the time dimension and accuracy gains of PLD is mostly from the depth dimension. However, without the temporal schedule enabled, the model diverges with *NaN* in the middle of half-precision (16-bit) training and has to switch to full-precision (32-bit) training, slowing down training speed. Furthermore, this concept of starting-easy and gradually increasing the difficulty of the learning problem has its roots in curriculum learning and often makes optimization easier. We adopt the temporal schedule since it is robust and helpful for training stability, retaining similar accuracy while reducing training cost considerably.

# 6  Conclusion

Unsupervised language model pre-training is a crucial step for getting state-of-the-art performance on NLP tasks. The current time for training such a model is excruciatingly long, and it is very much desirable to reduce the turnaround time for training such models. In this paper, we study the efficient training algorithms for pre-training BERT model for NLP tasks. We have conducted extensive analysis and found that model architecture is important when training Transformer-based models with stochastic depth. Using this insight, we propose the Switchable-Transformer block and a progressive layer-wise drop schedule. Our experiment results show that our training strategy achieves competitive performance to training a deep model from scratch at a faster rate.

## Broader Impact

Pre-training large-scale language models like BERT have an incredible ability to extract textual information and apply to a variety of NLP tasks, but pre-training requires significant compute and time. Pre-training the BERT baseline model is typically done through hardware acceleration and scaling the training on 100s to 1000s of GPUs across multiple nodes. However, such a method is very costly and consumes magnitudes higher energy.

The proposed solution achieves similar or better quality with shorter training time. It improves robustness to further reduce the hyperparameter tuning required, improving the productivity of AI scientists. It also saves hardware resources and trims down the total energy cost of in-situ, resource-constrained training, yielding a less amount of carbon footprint produced. Furthermore, the optimizations not only benefit BERT; they are also applicable to many other recent models such as RoBERTa [2], GPT-2 [9], XLNet [1], and UniLM [13], which all adopt Transformer as the backbone. Finally, our techniques can also help advance language understanding and inference, enabling enterprise or consumer-facing applications, such as conversational AI. We will open-source the code so that other practitioners and researchers can reproduce our results or re-use code into their ventures in this field.

There are no apparent negative outcomes. However, like other AI technology, we should be mindful of using it to transform our systems to be more efficient in fulfilling goodwill.

## Acknowledgments and Disclosure of Funding

The authors are grateful for the useful discussion with Xiaodong Liu, and Wenhan Wang. The authors appreciate the anonymous NeurIPS reviewers for providing constructive feedback for improving the quality of this paper. All authors are not funded by any other agency.

## Footnotes

[1]Appendix A provides detailed training hyperparameters.

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
