[Supplementary Material]

# A Pre-training Hyperparameters

Table 4 describes the hyperparameters for pre-training the baseline and PLD.

Table 4: Hyperparameters for pre-training the baseline and PLD.

| Hyperparameter | Baseline | PLD |
|---|---|---|
| Number of Layers | 12 | 12 |
| Hidden zies | 768 | 768 |
| Attention heads | 12 | 12 |
| Dropout | 0.1 | 0.1 |
| Attention dropout | 0.1 | 0.1 |
| Total batch size | 4K | 4K |
| Train micro batch size per gpu | 16 | 16 |
| Optimizer | Adam | Adam |
| Peak learning rate | 1e-04 | 1e-03 |
| Learning rate scheduler | warmup_linear_decay_exp | warmup_linear_decay_exp |
| Warmup ratio | 0.02 | 0.02 |
| Decay rate | 0.99 | 0.99 |
| Decay step | 1000 | 1000 |
| Max Training steps | 200000 | 200000 |
| Weight decay | 0.01 | 0.01 |
| Gradient clipping | 1 | 1 |

# B Establishing Identity Mapping with PreLN

Prior studies [28, 45] suggest that establishing *identity mapping* to keep a *clean* information path (no operations except addition) is helpful for easing optimization of networks with residual connections. With the change of PreLN, we can express the output of the i-th Transformer layer as the input $x_i$ of that layer plus a residual transformation function $f_{RT} = f_{S-ATTN}(f_{LN}(x_i)) + f_{FFN}(f_{LN}(x_i^{'}))$, and the output layer $x_L = x_l + \sum_{i=l}^{L-1} f_{RT}(x_i)$ as the recursive summation of preceding $f_{RT}$ functions in shallower layers (plus $x_l$). If we denote the loss function as $\mathcal{E}$, from the chain rule of backpropagation [46] we have:

$$\frac{\partial \mathcal{E}}{\partial x_l} = \frac{\partial \mathcal{E}}{\partial x_L} \frac{\partial x_L}{\partial x_l} = \frac{\partial \mathcal{E}}{\partial x_L}(1 + \frac{\partial}{\partial x_l} \sum_{i=l}^{L-1} f_{RT}(x_i)) \tag{5}$$

Eqn. 5 indicates that the gradient $\frac{\partial \mathcal{E}}{\partial X_l}$ can be decomposed into two additive terms: a term of $\frac{\partial \mathcal{E}}{\partial X_L}$ that propagates information directly back to any shallower $l$-th block without concerning how complex $\frac{\partial}{\partial x_l} \sum_{i=l}^{L-1} f_{RT}(x_i))$ would be, and another term of $\frac{\partial \mathcal{E}}{\partial X_L}(\frac{\partial}{\partial X_l} \sum_{i=l}^{L-1} f_{RT}(X_i))$ that propagates through the Transformer blocks. The equation also suggests that it is unlikely for the gradient $\frac{\partial}{\partial X_l}$ to be canceled out for a mini-batch, and in general the term $\frac{\partial}{\partial X_l} \sum_{i=l}^{L-1} f_{RT}(X_i)$ cannot be always -1 for all samples in a mini-batch. This explains why the gradients of Transformer layers in Fig. 1 become more balanced and do not vanish after identity mapping reordering. In contrast, the PostLN architecture has a series of layer normalization operations that constantly alter the signal that passes through the skip connection and impedes information propagation, causing both vanishing gradients and training instability. Overall, PreLN results in several useful characteristics such as avoiding vanishing/exploding gradient, stable optimization, and performance gain.

# C PreLN From the View of Unrolled Iterative Refinement

From a theoretical point of view [40], a noisy estimate for a representation by the first Transformer layer should, on average, be correct even though it might have high variance. The unrolled iterative refinement view says if we treat "identity mapping" (as in PreLN) as being an unbiased estimator for the target representation, then beyond the first layer, the subsequent Transformer layer outputs $x_i^n$ (e.g., $i \in 2...L$) are all estimators for the same latent representation $H^n$, where $H^n$ refers to the (unknown)

value towards which the $n$-th representation is converging. The unbiased estimator condition can then be written as the expected difference between the estimator and the final representation:

$$\mathop{\mathbb{E}}_{x \in X}[x_i^n - H^n] = 0 \tag{6}$$

With the PreLN equation, it follows that the expected difference between outputs of two consecutive layers is zero, because

$$\mathbb{E}[x_i^n - H^n] - \mathbb{E}[x_{i-1}^n - H^n] = 0 \Rightarrow \mathbb{E}[x_i^n - x_{i-1}^n] = 0 \tag{7}$$

If we write representation $x_i^n$ as a combination of $x_{i-1^n}$ and a residual $f_{RT}^n$, it follows from the above equation that the residual has to be zero-mean:

$$x_i^n = x_{i-1}^n + f_{RT}^n \Rightarrow \mathbb{E}[f_{RT}^n] = 0 \tag{8}$$

which we have empirically verified to be correct, as shown in Figure 2. Therefore, PreLN ensures that the expectation of the new estimate will be correct, and the iterative summation of the residual functions in the remaining layers determines the variance of the new estimate $\mathbb{E}[F_{RT\,i}]$.

# D  Downstream Task Accuracy Result

Fig. 10 shows the full comparison of the baseline and PLD, fine-tuned at different checkpoints. Overall, we observe that PLD not only trains BERT faster in pre-training but also preserves the performance on downstream tasks. Interestingly, our model achieves higher performance on MNLI, QNLI, QQP, RTE, SST-2, and CoLA on later checkpoints, indicating that the model trained with our approach also generalizes better on downstream tasks than our baseline does. From a knowledge transferability perspective, the goal of training a language model is to learn a good representation of natural language that ideally ignores the data-dependent noise and generalizes well to downstream tasks. However, training a model with a constant depth is at least somewhat noisy and can bias the model to prefer certain representations, whereas PLD enables more sub-network configurations to be created during training Transformer networks. Each of the L ST blocks is either active or inactive, resulting in $2^L$ possible network combinations. By selecting a different submodular in each mini-batch, PLD encourages the submodular to produce good results independently. This allows the unsupervised pre-training model to obtain a more general representation by averaging the noise patterns, which helps the model to better generalize to new tasks. On the other hand, during inference, the full network is presented, causing the effect of ensembling different sub-networks.

**The effect of learning rates on downstream tasks.**   We focus on evaluating larger datasets and exclude very small datasets, as we find that the validation scores on those datasets have a large variance for different random seeds.

For fine-tuning models on downstream tasks, we consider training with batch size 32 and performing a linear warmup for the first 10% of steps followed by a linear decay to 0. We fine-tune for 5 epochs and perform the evaluation on the development set. We report the median development set results for each task over five random initializations, without model ensemble.

Results are visualized in Fig. 11, which shows that the baseline is less robust on the choice of learning rates. Specifically, the fine-tuning results are often much worse with a large learning rate. In comparison, PLD is more robust and often achieves better results with large learning rates.

# E  Additional Results

Figure 10: The fine-tuning results at different checkpoints.

| | Baseline | PLD |
|---|---|---|
| 1.0e-05 | 0.817 | 0.815 |
| 3.0e-05 | 0.818 | 0.828 |
| 5.0e-05 | 0.808 | 0.829 |
| 7.0e-05 | 0.801 | 0.830 |
| 9.0e-05 | 0.789 | 0.831 |
| 1.0e-04 | 0.781 | 0.832 |

(a) MNLI-m

| | Baseline | PLD |
|---|---|---|
| 1.0e-05 | 0.822 | 0.819 |
| 3.0e-05 | 0.822 | 0.833 |
| 5.0e-05 | 0.814 | 0.828 |
| 7.0e-05 | 0.803 | 0.836 |
| 9.0e-05 | 0.793 | 0.836 |
| 1.0e-04 | 0.785 | 0.831 |

(b) MNLI-mm

| | Baseline | PLD |
|---|---|---|
| 1.0e-05 | 0.870 | 0.857 |
| 3.0e-05 | 0.866 | 0.875 |
| 5.0e-05 | 0.852 | 0.878 |
| 7.0e-05 | 0.836 | 0.880 |
| 9.0e-05 | 0.822 | 0.878 |
| 1.0e-04 | 0.811 | 0.875 |

(c) QNLI

| | Baseline | PLD |
|---|---|---|
| 1.0e-05 | 0.905 | 0.887 |
| 3.0e-05 | 0.906 | 0.905 |
| 5.0e-05 | 0.904 | 0.908 |
| 7.0e-05 | 0.898 | 0.908 |
| 9.0e-05 | 0.632 | 0.909 |
| 1.0e-04 | 0.632 | 0.910 |

(d) QQP

| | Baseline | PLD |
|---|---|---|
| 1.0e-05 | 0.664 | 0.643 |
| 3.0e-05 | 0.632 | 0.650 |
| 5.0e-05 | 0.664 | 0.679 |
| 7.0e-05 | 0.527 | 0.657 |
| 9.0e-05 | 0.527 | 0.690 |
| 1.0e-04 | 0.527 | 0.690 |

(e) RTE

| | Baseline | PLD |
|---|---|---|
| 1.0e-05 | 0.907 | 0.911 |
| 3.0e-05 | 0.903 | 0.915 |
| 5.0e-05 | 0.892 | 0.919 |
| 7.0e-05 | 0.882 | 0.914 |
| 9.0e-05 | 0.866 | 0.911 |
| 1.0e-04 | 0.806 | 0.914 |

(f) SST-2

Figure 11: The fine-tuning results at different checkpoints.

Figure 12: Convergence curves varying the keep ratio $\bar{\theta}$.