[Reviews · NeurIPS 2020]

Review 1

Summary and Contributions: This paper proposes to accelerate training of Transformer networks by progressively reducing Transformer layers from the network during training. First, it compares two different architectures of BERT, PostLN and PreLN. PostLN applies layer normalization after the element-wise addition in Transformer blocks. The PreLN changes the placement of the location of layer normalization by placing it only on the input stream of the sublayers. It finds that PostLN is more sensitive to the choice of hyperparameters, and training often diverges with more aggressive learning rates whereas PreLN avoids vanishing gradients and leads to more stable optimization. Second, it plots the L2 distance and cosine similarity of the input and output embeddings for BERT with PostLN and PreLN, at different layers and different steps. The plots show that as the training proceeds, although the dissimilarity remains relatively high and bumpy for PostLN, the similarity from PreLN starts to increase over successive layers, indicating that while PostLN is still trying to produce new representations that are very different across layers, the dissimilarity from PreLN is getting close to zero for upper layers, indicating that the upper layers are getting similar estimations. Furthermore, the plots indicate, since the lower layers remain to have a relatively high dissimilarity, they should be less frequently dropped. The usefulness of preLN in dropping layers is established by experimental results. When layers are dropped randomly with dropout=0.5, postLN significantly reduces performance in comparison to preLN. In particular, for each mini-batch, they decide whether a particular layer (connected by element-wise addition) is activated or not. They use a progressively increasing layer dropout probability across the network. Moreover, lower layers have lower drop probability linearly scaled by their depth. Their method achieves a 24% time reduction when training the same number of samples, and also converges faster in terms of validation loss. The combination of these two factors, yields 2.8× speedup in end-to-end wall-clock training time over the baseline. Further, they show that downstream task performance remains the same when BERT is trained with their approach. Note that, theirs method only affects the training procedure, architecture and inference remains the same.

Strengths: A practically useful method to reduce BERT training time. The approach has been well investigated empirically. The experiments showing the effect of preLN and postLN are quite informative and provide insights into the similarity of layer input and output for top-vs-bottom transformer layers. These insights justify their approach of progressively dropping layers and dropping top layers with higher probability than the bottom layers.

Weaknesses: It is completely an empirical work with no theoretical underpinnings. The innovative contribution is limited as dropout is a standard procedure. However, they show that stochastic dropout in BERT training performs significantly worse and their proposed systematic dropout performs similar/better than the original no-dropout BERT training.

Correctness: Yes.

Clarity: Well written.

Relation to Prior Work: Yes.

Reproducibility: Yes

Additional Feedback: Thanks for the rebuttal. I went through the rebuttal. My assessment of the paper remains the same.


Review 2

Summary and Contributions: The paper aims to reduce the extensive computational cost of training transformer language models. The key method is to use layer dropping by leveraging the characterization of post / previous layers normalization and lower / higher layers. This acceleration scheme does not affect the overall performance of the pre-trained language model. The method achieves good performance to training a deep model from scratch at a faster rate.

Strengths: This issue is very important especially for researchers with limited computational resources. The analysis of the representation across layers is very insightful. The solution is easy to understand and seems powerful enough. The experiment shows the scalability of this approach on the SOTA language model across 64-GPU. The ablation also shows the speedup and is not limited to a particular learning rate. The author says that open-source the code in board impact.

Weaknesses: 1) One of the major challenges of training transformers is its large memory footprint (GPU memory quickly used up). By doing layer dropping, it seems the footprint is not reduced. I suggest trying to re-implement the design using tensorflow and combine the algorithm with model-parallelism to reduce memory footprint and train it using larger batch-size. 2) Can this method be generalized to some light-weight language tasks on single GPU training? The benchmarks are tested on 64-GPU which are not common training platform for researchers. 3) The analysis of how stable it is to leverage the PLD method is missing. Does PLD achieve the same std as the baseline training or it would destabilize the training accuracy across multiple experiments? Minor: Table 1 row 2 should be 29.22h.

Correctness: Yes.

Clarity: Good.

Relation to Prior Work: Yes.

Reproducibility: Yes

Additional Feedback: See weakness for suggestions and questions. Update: I will keep my score to 7. I think the experiments in this paper are very thorough. I suggest the author to clarify the related works that are missing mentioned by R4.


Review 3

Summary and Contributions: The paper summarizes various stability-related observations on multiple aspects of a deep transformer and proposed a useful method for accelerate large-scale training of the transformer models (named PLD) that is able to significantly improve both the efficiency and the stability of the model. While closely related to some other works (e.g., those that study the pre- and post-LN properties in the transformer), this paper is able to exploit these phenomenons and designed a useful regularization technique that non-uniformly drop the layers.

Strengths: The paper is well-written, well-motivated, and has provided a convincing set of empirical evaluations that supported its claim. Unlike the prior works on accelerating the transformer model, the methodology at the center of this paper is actually empirically driven by careful observations on the training dynamics of a large-scale transformer. The method is both simple and effective, as the authors have demonstrated on large-scale models and tasks.

Weaknesses: 1) The architectural contribution of this approach is in my view still minimal. For example, the difference between pre- and post-LN methods have been well-studied [1], including a similar kind of observation on the gradient norm change across the layers. I therefore would be reluctant to make the "identity mapping reordering" as a novel contribution of the "switchable transformer block". Architectural, what is proposed is simply to add a switch to each transformer block. Such adaptive computation in deep learning is not new; for example, see [2]. This work only provides a schedule to the layer dropping, which makes me feel has limited novelty in itself (but is important to the community!). 2) It's still unclear to me whether some of the empirical observations made here (e.g., the effect of lesioning) is a matter of dataset/domain or a more general phenomenon. For example, on applications without specific embedding, like pixel-generation/image transformers, would the transformer layer still have the diminishingly small iterative refinement? If so, does one need to adjust the schedule of the layer dropping accordingly? Therefore, I'm not sure how well the layer dropping schedule proposed here could generalize--- or is it just a BERT/NLP phenomenon. [1] https://arxiv.org/pdf/2002.04745.pdf [2] https://arxiv.org/pdf/1711.09485.pdf

Correctness: Yes, the claims are mostly well supported and the empirical methodology seems correct.

Clarity: Yes. (But please make the figures and their legends easier to see.)

Relation to Prior Work: Yes.

Reproducibility: Yes

Additional Feedback: Post-rebuttal: I've read the rebuttal from the authors and am generally satisfied with the author response (they missed some details due to the page limit). Thank you! But still, the architectural contribution of the propose approach is relatively minimal and mostly empirical--- and so I decided to keep my score at 6. ----------------------------------------------------------------------- 1) I found the beginning of Sec. 4.2 a bit confusing when I first read the paper. The schedule theta(t), as the paper currently says, is a temporal schedule on t. However, it can be easily mistaken to be the temporal dimension of the sequence (i.e., the sequence length dimension). I would rephrase this part to make it explicit that you mean the training step. 2) Have you tried different scheduling? The discussion on polynomial schedules are great, but there are also other schedules that will eventually saturate at the \bar{\theta} target. How important is the schedule itself to the performance/stability of the model? What if you just set it to be a linear decay?


Review 4

Summary and Contributions: In this paper the authors propose an approach for faster and more stable training of Transformer-based language models. They do this by allowing the model to drop certain layers, according to a schedule throughout training, such that the expected depth of the netowork increases throughout training. They show that this allows for ~2.5x speed up in pretraining, and better stability in fine-tuning given different hyperparameters. Following author response: Given the detailed response from the authors and reading the other reviews (e.g. others followed the paper better than I), I've increased my score to a 4. My main concern is lack of comparison to previous work, particularly LayerDrop. I think their review of previous work is completely insufficient (as demonstrated by the lack of a related work section). Given that there has been so little work on accelerating BERT training, I feel that it should all be discussed, even if direct empirical comparison is arguably not necessary, such as the ELECTRA paper. Instead, none was discussed, including highly relevant work that could serve as a baseline (e.g. LayerDrop). I agree that the paper has merits and it's an interesting contribution, but I can't recommend that it be accepted in it's current form.

Strengths: The proposed approach seems to achieve faster training than a baseline BERT, maintaining comparable (development set) scores on GLUE while training ~2.5x faster. They also claim that fine-tuning is more stable with this model, though I don't find the empirical results entirely convincing.

Weaknesses: - Comparison to related work is insufficient to be able to evaluate the novelty and potential impact of this work. There is no related work section, and no comparison to related work in empirical results, except the baseline BERT model. - Empirical results not thorough enough to be convincing. See correctness below.

Correctness: - the authors only compare to BERT and don't compare to other LM pretraining approaches, and don't compare to other approaches that claim to improve LM training time, such as ELECTRA (https://openreview.net/forum?id=r1xMH1BtvB) - It's unclear whether they use the same train/dev split as the original BERT, though they do compare to their own reimplementation, presumably trained on their data split. - It appears that no test set results are reported, only dev set. - Empirical results not thorough enough to be convincing. For example, in Figure 9, these are some of the largest and most stable GLUE datasets for fine-tuning. If your training approach increases stability (1) compare to simply moving the layer norm, which is also supposed to improve stability on its own, and has been done in previous work; (2) show that stability is improved on the smaller and less stable GLUE datasets. This is potentially a very interesting and impactful result (hyperparameter tuning uses a lot of compute!) but as of yet, the presented results are not convincing. - Similarly, I would like to see far more ablation of the approach. How does a more naive/simple schedule of increasing model depth perform? This is what I initially expected the paper to be. Do you really need sampling? - Do you really need the more complicated ST blocks?

Clarity: The paper has local coherence but lacks some global coherence, e.g. in section 3 I have no idea at that point why we're looking at stabilizing BERT training by analyzing pre- vs. post-LN. There's no foreshadowing of this in the abstract or introduction. In fact, I think the entire section 3 should really be merged into empirical results and drastically reduced to make room for more discussion of and comparison to related work. Overall, I think this paper needs to be substantially revised to be ready for publication.

Relation to Prior Work: No; this paper has no related work section. Some prior work is cited throughout, but his paper is missing citations of at least two highly related works, e.g. LayerDrop (https://arxiv.org/abs/1909.11556) and layer reordering (https://arxiv.org/abs/1911.03864), and likely more. There is essentially no comparison to prior work in empirical results, only the most straightforward baseline (original BERT model).

Reproducibility: Yes

Additional Feedback: - many figures are quite small, requiring me to zoom in substantially to see, and they are not in a lossless format (which makes them even harder to see without zooming). - regarding your statement about dropping sublayers, see: https://arxiv.org/abs/1909.11556 and https://arxiv.org/abs/1911.03864 - citations don't appear to be the same color as in the normal NeurIPS format; make sure you're using the correct style file / paper format. it also looks like you've used negative vspace on section headings and around figures/tables (e.g. Section 6 (conclusion), and Table 2) which I don't believe is allowed, as it's unfair to others who trimmed the text of their papers rather than cheating the formatting, and it makes the paper harder to read. There are many places you could reduce the text in your paper instead, for example under "Datasets" in Section 5, you could remove much of that text, as I assume your data setup is the same as that of Devlin et al. - recommend removing excess rules (e.g. vertical rules and many horizontal ones) in tables to make them more readable. you don't need a line around every single cell. - rather than listing the checkpoint number it would be more useful to list the wall-clock training time, in table 2


Review 5

Summary and Contributions: This paper proposes to use two techniques to accelerate the training of BERT. The first is to use PreLN instead of PostLN. The second is to use progressive layer dropping (PLD). Progressive layer dropping will use a time-dependent and depth-dependent schedule to choose a set of layers to drop. Experimental results on both masked language modeling pre-training and GLUE fine-tuning show the effectiveness of this method. ========================== I have read the authors' response and appreciate their effort.

Strengths: 1. Extensive experiments on BERT show that PLD allows the pre-training to be 2.5× faster than the baseline to get a similar accuracy on downstream tasks. This is impressive. 2. The analysis in "Corroboration of Training Dynamics" is interesting and well motivates the proposed PLD method. The "unrolled iterative refinement" hypothesis is verified for BERT in Figure 4. 3. The description of PLD and its motivation is clear.

Weaknesses: 1. A necessary baseline is missing in the ablation study: PreLN BERT without PLD. Without comparing with this baseline, it is unclear whether the 2.5x speedup comes from PreLN or from PLD. 2. It would be better if the paper could try a few more schedules in the ablation study for "Progress along the time dimension" (section 4.2), such as the linear decay schedule. I believe this would make the paper more complete.

Correctness: The empirical methodology seems to be correct. The implementation details are very clear.

Clarity: The paper is well motived and well written.

Relation to Prior Work: The discussion is clear.

Reproducibility: Yes

Additional Feedback: Comments: 1. Although the discussion of "Corroboration of Training Dynamics" (Section 3.2) is interesting, I believe there is no need to analyze the norm of the gradients for PreLN again in this paper (Section 3.1, Figure 1). The discussion in this paper is already included in previous work [30-34].

[Author Response · NeurIPS 2020]

**Q1:** Both reviewer #4 and reviewer #5 think it is essential to compare the proposed method with Pre-LayerNorm.

**A:** We added additional experiments to investigate the question on how PLD compares with PreLN? We tried both Pre-LN with the hyperparameters used for training Post-LN (lr=1e-4) and the hyperparameters used for PLD training (lr=1e-3) to address the effect from the choice of hyperparameters. We trained all configurations for the same number of epochs and finetuned following the standard procedure. In both cases, PreLN is 24% slower than PLD, because PreLN still needs to perform the full forward and backward propagation in each iteration. The table below shows the finetuning results on GLUE tasks. When trained with the same hyperparameters as Post-LN, Pre-LN appears to have a much worse GLUE score (80.2) compared with Post-LN (82.1) on downstream tasks. This is because Pre-LN restricts layer outputs from depending too much on their own residual branches and inhibits the network from reaching its full potential, as recently studied in https://arxiv.org/pdf/2004.08249.pdf. When trained with the large learning rate as PLD, PreLN's result have improved to 82.6 but is 0.6 points worse than PLD (83.2), despite using 24% more compute resource. PLD achieves better accuracy than PreLN because it encourages each residual branch to produce good results independently.

| Model | RTE (Acc.) | MRPC (F1/Acc.) | STS-B (PCC/SCC) | CoLA (MCC) | SST-2 (Acc.) | QNLI (Acc.) | QQP (F1/Acc.) | MNLI-m/mm (Acc.) | GLUE |
|---|---|---|---|---|---|---|---|---|---|
| BERT (Original) | 66.4 | **88.9**/84.8 | 87.1/89.2 | 52.1 | **93.5** | **90.5** | 71.2/89.2 | **84.6/83.4** | 80.7 |
| BERT + PostLN | 67.8 | 88.0/86.0 | 89.5/89.2 | 52.5 | 91.2 | 87.1 | 89.0/90.6 | 82.5/83.4 | 82.1 |
| BERT + PreLN + Same lr | 66.0 | 85.9/83.3 | 88.2/87.9 | 46.4 | 90.5 | 85.5 | 89.0/90.6 | 81.6/81.6 | 80.2 |
| BERT + PreLN + Large lr | 67.8 | 86.7/84.5 | **89.6/89.1** | 54.6 | 91.9 | 88.1 | 89.3/**90.9** | **83.6/83.7** | 82.6 |
| Shallow BERT + PreLN + Large lr | 66.0 | 85.9/83.5 | 89.5/88.9 | 54.7 | 91.8 | 86.1 | 89.0/90.6 | 82.7/82.9 | 81.8 |
| BERT + PreLN + Large lr + Random | 68.2 | 88.2/86.2 | 89.3/88.8 | 56.8 | 91.5 | 87.2 | 88.6/90.3 | 82.9/83.3 | 82.7 |
| BERT + PreLN + Large lr + TD only | 68.2 | 88.6/**86.7** | 89.4/88.9 | 55.9 | 91.3 | 86.8 | 89.1/90.7 | 82.7/83.1 | 82.7 |
| BERT + PreLN + Large lr + PLD (TD + DD) | **69.0** | **88.9**/86.5 | **89.6/89.1** | **59.4** | 91.8 | 88.0 | **89.4/90.9** | 83.1/83.5 | **83.2** |

**Q2:** Reviewer #3, #4, #5 ask about a comparison to simpler and alternative schedules.

**A:** The current schedule is actually simple. Let us elaborate it in a more intuitive way. Along the temporal dimension (TD), we use the standard exponential decay schedule. This is because there are large embedding shifts at the beginning of the training. Intuitively, we do not want to perturb it with layer drop that can destabilize the training. Along the depth dimension (DD), we use a simple linear decay schedule from $p_0 = 1$ for the first Transformer layer, to $p_L$ for the last Transformer layer. This is because upper layers are often getting similar estimations, especially in the later phase of the training. Our schedule induces an adaptive regularization scheme that smoothly increases the difficulty of sampling. This concept of starting-easy and gradually increasing the difficulty of the learning problem has its roots in curriculum learning (https://icml.cc/Conferences/2009/papers/119.pdf) and allows one to train better models.

We have since run more ablation experiments to evaluate the effectiveness of PLD and added results in Table 1. (1) **Shallow BERT + PreLN + Large lr** directly trains a 9-layer BERT without sampling. This configuration underperforms PreLN by 0.8 points and is 1.4 points worse than PLD likely because the model capacity has been reduced by the loss of parameters. (2) **BERT + PreLN + Large lr + Random** drops layers randomly with a fixed ratio (i.e., it has the same compute cost but without any schedule). It is 0.9 points better than shallow BERT under the same compute cost and 0.1 points better than PreLN while being 24% faster, indicating the strong regularization effect from stochastic depth. It is 0.5 points worse than PLD, which demonstrates the benefit of having schedules enabled. (3) **BERT + PreLN + Large lr + TD only (32-bit*)** disables the DD schedule in training. Its GLUE score matches "Random", suggesting that the exponential decay schedule (TD) has similar performance as the fixed constant schedule along the temporal dimension and accuracy gains of PLD is mostly from the DD schedule. However, without the temporal schedule enabled, the model diverges with NaN in the middle of half-precision (16-bit) training and has to switch to full-precision (32-bit) training, slowing down training speed. We adopted the temporal schedule since it is robust and helpful for training stability, retaining similar accuracy while reducing training cost considerably.

**Q3**: Reviewer #2 and #3 asks if this method be generalized to other tasks? **A:** We currently work on BERT-like model training. But we also think the techniques are likely generalizable across other workloads if they also exhibit the unrolled iterative refinement phenomenon. This would make an interesting future study. ‖ **Q4**: Reviewer #2 asks whether PLD achieves the same std as the baseline training across multiple experiments? **A:** PLD achieves similar std as the baseline across multiple runs. Fig. 13 in the Appendix reports the median and std of finetuning results for all GLUE tasks. ‖ **Q5**: Reviewer #4 asks to show the stability is improved on the smaller and less stable GLUE dataset. **A:** Fig. 13 and Fig.14 in the Appendix have more GLUE results. For example, Fig. 13 and Fig.14 show results on RTE, the smallest dataset in GLUE. Overall, for small datasets such as CoLA, MRPC, and RTE, PLD is more stable and outperforms the baseline with a statistically significant difference. ‖ **Q6**: Reviewer #4 asks about the comparison with ELECTRA. **A:** we think these are two complementary techniques to speedup pre-training. ELECTRA speeds up the pre-training by replacing masked tokens with alternatives sampled from a generator framework and training a discriminator to predict the replaced token, whereas we speed up the pre-training speed through an architectural change and a schedule for layer dropping. Working together, they could reduce the end-to-end pre-training time even further.

[Meta-Review · NeurIPS 2020]

The proposed method for training BERT is practically useful. My main concern on this paper is that the novelty in this paper is somewhat limited. It combines two existing techniques. One is PreLN which has been well studied in the literature for training BERT, and the other is stochastically dropping layers which was first proposed for training CV models. On the other hand, how to effectively combine these two techniques and fine tune to make them work for training BERT needs certain efforts. The provided training recipe should be interesting for practitioners.